# The energetic basis for smooth human arm movements

Jeremy D Wong[1]*, Tyler Cluff[1,2], Arthur D Kuo[1]

[1]University of Calgary, Faculty of Kinesiology, Department of Biomedical Engineering, Calgary, Canada; [2]Hotchkiss Brain Institute, University of Calgary, Calgary, Canada

**Abstract** The central nervous system plans human reaching movements with stereotypically smooth kinematic trajectories and fairly consistent durations. Smoothness seems to be explained by accuracy as a primary movement objective, whereas duration seems to economize energy expenditure. But the current understanding of energy expenditure does not explain smoothness, so that two aspects of the same movement are governed by seemingly incompatible objectives. Here, we show that smoothness is actually economical, because humans expend more metabolic energy for jerkier motions. The proposed mechanism is an underappreciated cost proportional to the rate of muscle force production, for calcium transport to activate muscle. We experimentally tested that energy cost in humans (N = 10) performing bimanual reaches cyclically. The empirical cost was then demonstrated to predict smooth, discrete reaches, previously attributed to accuracy alone. A mechanistic, physiologically measurable, energy cost may therefore explain both smoothness and duration in terms of economy, and help resolve motor redundancy in reaching movements.

## Editor's evaluation

This paper will be of interest to researchers in the fields of biomechanics, movement control, and decision making. A novel, mechanistic model of metabolic cost is presented to account for a phenomenon not explained by current models of metabolic energy. This is followed by a demonstration of how this metabolic model can improve our understanding of movement control by revealing an energetic basis for smooth movements.

*For correspondence:
jeremy.wong2@ucalgary.ca

**Competing interest:** The authors declare that no competing interests exist.

## Introduction

Upper extremity reaching movements are characterized by a stereotypical, bell-shaped speed profile for the hand's motion to its target (*Figure 1A*). The profile's smoothness seems to preserve kinematic accuracy (*Harris and Wolpert, 1998*) and have little to do with the effort needed to produce the motion. But effort or energy expenditure appear to affect other aspects of reaching (*Huang et al., 2012*; *Shadmehr et al., 2019*), and influence a vast array of other animal behaviors and actions (*Alexander, 1996*). It seems possible that effort or energy do influence the bell-shaped profile, but have gone unrecognized because of incomplete quantification of such costs. If so, then dynamic goals including effort could play a key role in movement planning.

The kinematic goal for accuracy may be expressed quantitatively as minimization of the final endpoint position variance (*Harris and Wolpert, 1998*). Non-smooth motions introduce inaccuracy because motor noise increases with motor command amplitude, a phenomenon termed signal-dependent noise (*Matthews, 1996*; *Sutton and Sykes, 1967*). It predicts well the speed profiles for not only the hand but also the eye. It explains why more curved or more accurate motions need to be slower, and also subsumes an older theory for minimizing kinematic jerk (*Flash and Hogan, 1985*). The single objective of movement variance explains multiple aspects of smooth movements,

**Figure 1.** Goal-directed reaching tasks and optimization criteria. (**A**) Typical experiments for point-to-point movements between targets. (**B**) Hand speed trajectories vs. time. Kinematic objectives such as minimizing jerk or variance predict the observed smooth, bell-shaped profiles for hand speed. (**C**) A number of effort-based objectives such as minimizing work or squared muscle force predict trajectories that are not smooth or not bell-shaped (*Nelson, 1983*).

and makes better predictions than competing theories (*Diedrichsen et al., 2010*; *Haith et al., 2012*; *Todorov, 2004*).

There are nonetheless reasons to consider effort. Many optimal control tasks must include an explicit objective for effort, without which movements would be expected to occur at maximal effort ('bang-bang control', *Harris and Wolpert, 1998*; *Bryson and Ho, 1975*). In addition, metabolic energy expenditure is substantial during novel reaching tasks and decreases as adaptation progresses (*Huang et al., 2012*). Such a cost also helps to determine movement duration and vigor (*Shadmehr et al., 2016*), not addressed by the minimum-variance hypothesis. Indeed, optimal control studies have long examined effort costs such as for muscle force (*Kolossiatis et al., 2016*), mechanical work (*Alexander, 1997*), squared force or activation (*Nelson, 1983*; *Ma et al., 1994*), or 'torque-change' (integral of squared joint torque derivatives; *Uno et al., 1989*). But many such costs produce non-smooth velocity profiles (*Figure 1B*), or lack physiological justification, or both. Some studies have included explicit models of muscle energy expenditure, but without testing such costs physiologically (*Kistemaker et al., 2010*). There is good evidence that energy expenditure is relevant to reaching (*Shadmehr et al., 2016*), but no physiologically tested cost function predicts the velocity profiles of reaching as well as the minimum variance hypothesis.

The issue could be that metabolic energy expenditure for muscle is not quantitatively well-understood. Energy is expended in proportion to force and time ('tension-time integral') in isometric conditions (*Crow and Kushmerick, 1982*), and in proportion to mechanical work in steady work conditions (*Barclay, 2015*; *Margaria, 1976*), neither of which apply well to reaching. There is, however, a less-appreciated cost for muscles that increases with brief bursts of intermittent or cyclic action. It is due to active calcium transport to activate/deactivate muscle, observed in both isolated muscle preparations (*Hogan et al., 1998*) and whole organisms (*Bergström and Hultman, 1988*). It has also been hypothesized quantitatively (*Doke and Kuo, 2007*), as a cost per contraction roughly proportional to the rate of change of muscle force. Such a cost has indeed been observed in a variety of lower extremity tasks (*Dean and Kuo, 2011*; *Doke et al., 2005*; *van der Zee and Kuo, 2020*). It has a mechanistic and physiological basis, is supported by experimental evidence, and would appear to penalize jerky motions due to their energetic cost. What is not known is whether this energetic cost can explain reaching.

We therefore tested whether there is an energetic basis for reaching movements (*Figure 2*). We did so by measuring oxygen consumption during steady-state cyclic reaching motions. The expectation was that the proposed force-rate cost would cost metabolic energy in excess of what could be explained by mechanical work. We next applied the empirically derived cost for both force-rate and work to an optimal control model of discrete, point-to-point reaching, and tested whether it could predict the smooth, bell-shaped velocities normally attributed to minimum-variance. If the proposed cost is observed as expected and predicts bell-shaped profiles, it could potentially provide

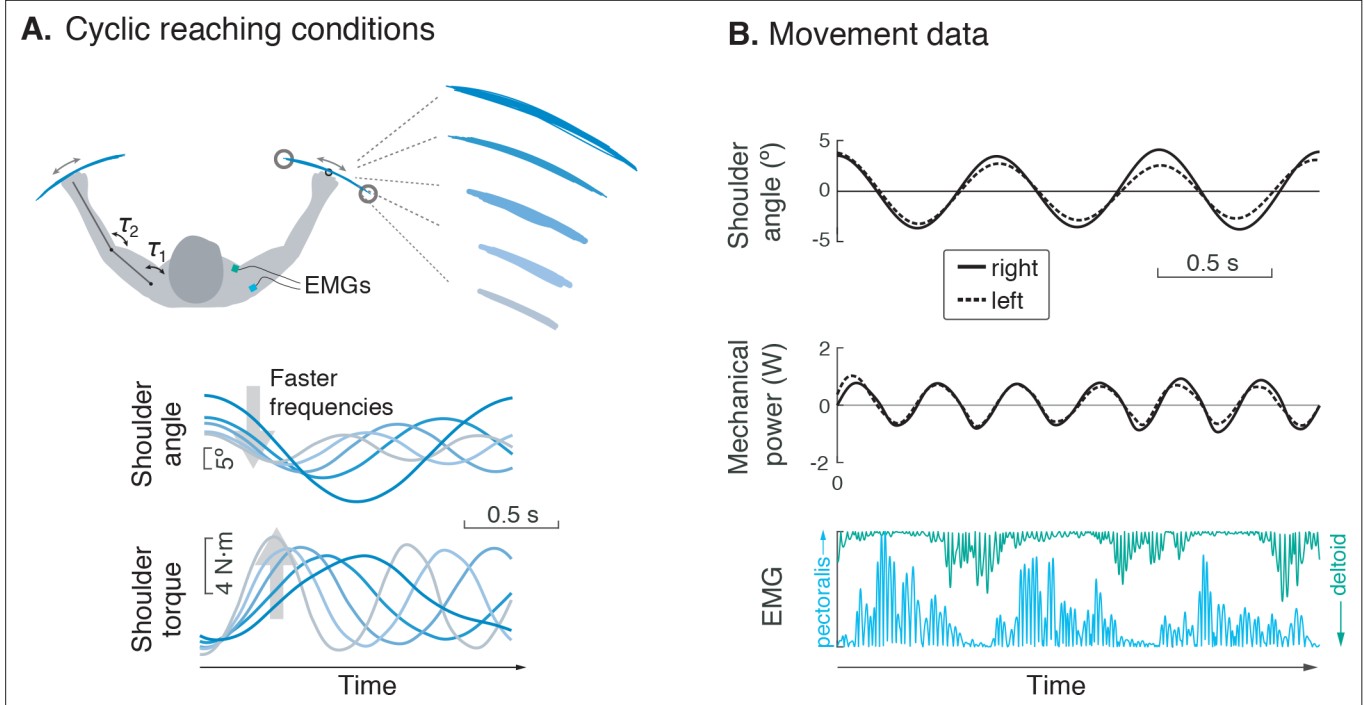

**Figure 2.** Experiment for metabolic cost of cyclic reaching. (**A**) Cyclic reaching was performed bimanually and symmetrically in the horizontal plane, primarily about the shoulders. To isolate the hypothesized force-rate cost from the energetic cost of work, movements were varied to yield fixed mechanical power, by decreasing amplitudes with increasing movement frequency. (**B**) Movement data included shoulder angle, mechanical power, electromyography (EMG), and (not shown) metabolic energy expenditure via expired gas respirometry.

a re-interpretation of existing theories based on kinematics alone, and integrate energy expenditure into a general framework for planning reaching movements.

## Results

Model optimization resulted in a prediction of the metabolic cost of cyclic reaching movements. With movement amplitude decreasing with movement frequency, metabolic cost was predicted to increase with movement frequency $f$ to the 5/2 power (***Figure 3A***). This is in proportion to the force-rate cost, also expected to increase with $f^{5/2}$. We also expected a fixed metabolic cost for mechanical work, because these conditions result in fixed mechanical power across frequencies. The specific movement conditions needed to separate the costs of work and force-rate were as follows: movement amplitude decreasing according to $f^{-3/2}$ (***Figure 3B***), joint torque increasing with $f^{1/2}$ (***Figure 3C***), and hand speed decreasing with $f^{-1/2}$ (***Figure 3D***). Thus, even though mechanical power is expected to contribute substantially to metabolic cost, the force-rate cost can be tested for an increasing contribution to overall metabolic cost.

We found that the rate of metabolic energy expenditure increased substantially with movement frequency, even as the rate of mechanical work was nearly constant. We first confirmed that cyclic reaching was performed largely by sinusoidal motions at the shoulder, across all conditions (***Figure 4***). These were accompanied by approximately sinusoidal torque and power, and fairly consistent EMG profiles. Under such conditions, subjects expended more than triple (a factor of 3.56) the net metabolic power for about twice the frequency (a factor of 2.33), with 5.32 ± 2.73 W at the lowest frequency of 0.58 Hz, compared to 18.95 ± 6.02 W at the highest frequency of 1.36 Hz (***Figure 5A***). As predicted, metabolic rate increased approximately with $f^{5/2}$ (***Equation 9***; adjusted $R^2$ = 0.50; p = 1e-8; ***Figure 4a***; ***Table 1***).

Other aspects of the cyclic reaching task were as prescribed and intended (***Figure 5B–E***; ***Table 1***). Reach amplitudes decreased according to the targets, approximately with $f^{-3/2}$ (***Figure 5B***). Shoulder torque amplitude and endpoint speed also changed with respectively $f^{1/2}$ (***Figure 5C***; adjusted $R^2$ =

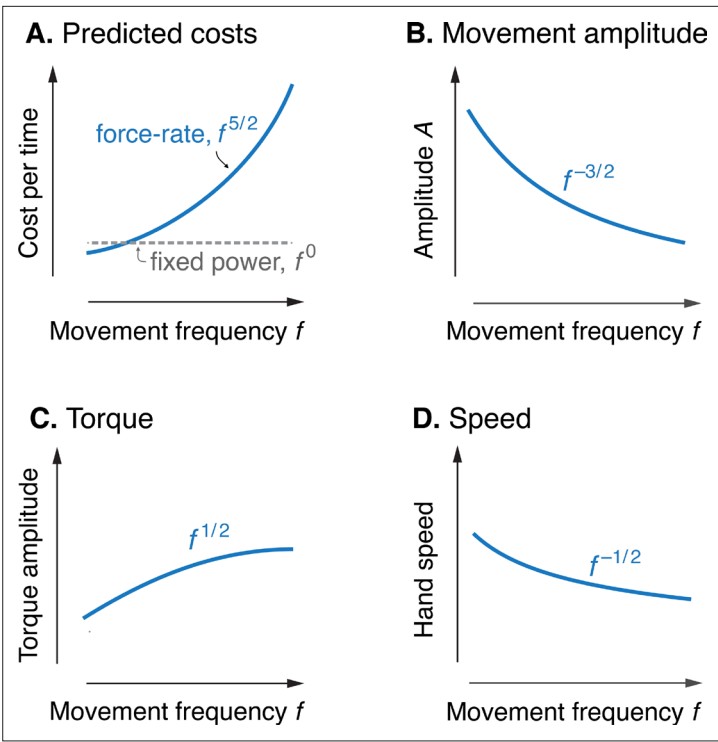

**Figure 3.** Predicted cost and dynamics for cyclic reaching, as a function of movement frequency $f$. (**A**) Force-rate cost is predicted to increase with $f^{5/2}$, whereas cost for mechanical work is predicted to remain constant for fixed power conditions. (**B**) Fixed power is achieved by specifying movement amplitude $A$ to decrease with frequency, according to $f^{-3/2}$. (**C**) Torque amplitudes are expected to increase modestly, with $f^{1/2}$. (**D**) Peak hand speed is expected to decrease, with $f^{-1/2}$.

0.52; p = 4e-9) $f^{-1/2}$ (*Figure 5D*; $R^2$ = 0.93; p = 7e-29). Consistent with the fixed-power condition, average positive mechanical power did not change significantly with frequency $f$ (*Figure 5E*; slope = 0.081 ± 0.13 W.s⁻¹; mixed-effects linear model with a fixed effect proportional to $f^1$, and individual subject offsets as random effects; p = 0.16). Amplitude of torque rate per time increased more sharply, approximately with $f^{5/2}$ (*Figure 5E*), with coefficient $b$ of 78.93 ± 6.55 CI, 95% confidence interval.

The net metabolic cost was also consistent with the hypothesized sum of separate terms for positive mechanical work and force-rate (*Figure 6*). This is demonstrated with metabolic power as a function of movement frequency $f$, and as a function of force-rate per time. With positive mechanical work at a fixed rate of about 1.2 W, the metabolic cost of work was expected to be constant at approximately 5 W. The difference between net metabolic rate and the constant work cost yielded the remaining force-rate metabolic power, increasing approximately with $f^{5/2}$ (*Figure 6A*). This same force-rate cost could also be expressed as a linear function of the empirical torque rate per time, with an estimated coefficient of $c_t$ = 8.5e-2 (*Figure 6B*; see *Equation 11*); joint torque is treated as proportional to muscle force, assuming constant shoulder moment arm. In terms of proportions, mechanical power accounted for about 94% of the net metabolic cost at 0.58 Hz, and 26% at 1.36 Hz. Correspondingly, force-rate accounted for about 6% and 74% of net metabolic rate at the two respective frequencies.

Muscle EMG amplitudes increased with movement frequency (*Figure 7*). Deltoid and pectoralis both increased approximately with $f^{3/2}$ (pectoralis: $R^2$ = 0.65; p = 1.1e-6; deltoid: $R^2$ = 0.56; p = 1.5e-5), as did the co-contraction index ($R^2$ = 0.58; p = 0.0009). This was consistent with expectations of muscle activation increasing faster than torque for increasing movement frequencies.

## Cross-validation of metabolic cost during cyclic reaching

Separate cross-validation trials agreed well with force-rate coefficients. The second group of subjects moved with slightly increasing mechanical power, and slightly higher metabolic cost (*Figure 8*). But

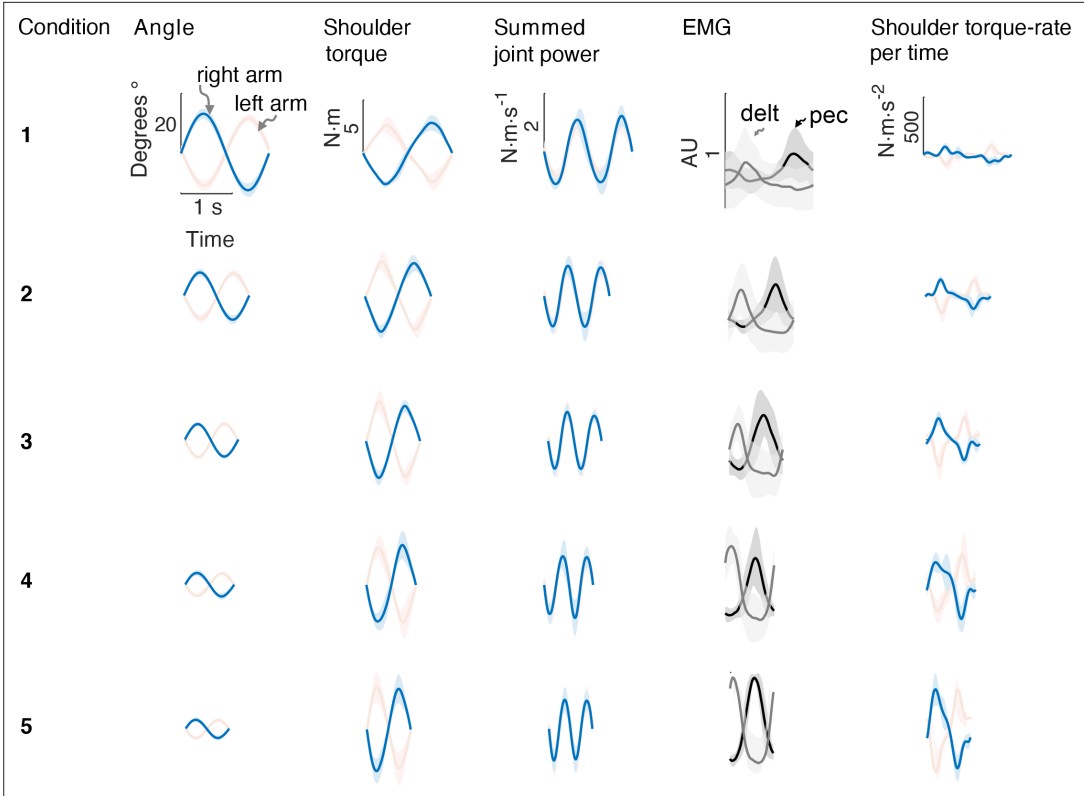

**Figure 4.** Average bimanual reach trajectories and EMG. Mean (± s.d.; $N$=5 EMG subjects) angular displacement, shoulder torque, summed joint power, EMG, and torque rate-per-time for five conditions from Experiment 1. Right (blue) and left (red) arms are shown separately.

applying the cost coefficient $c_t$ derived from the primary experiment, the model (*Equations 1; 10*) was nevertheless able to predict cross-validation costs reasonably well ($R^2$ = 0.42; p = 2.7e-6).

## Passive elastic energy storage during cyclic reaching

The estimated natural frequency of cyclic arm motions was 2.83 ± 0.56 Hz. This suggests a rotational stiffness about the shoulder joint of about 250 N·m·rad$^{-1}$, if series elasticity were assumed for shoulder muscles. With passive elastic energy storage, the average positive mechanical power of muscle fascicles would decrease slightly, from about 0.5 W per arm to 0.33 W. Thus, series elasticity would cause active mechanical power to decrease with movement frequency, as energy expenditure increased.

## Hill-type model does not predict experimentally observed energy cost

The Hill-type model's predicted net energy cost increased approximately linearly with movement frequency, from 33 W to 47 W. The model dramatically over-predicted the net metabolic cost for all movements (by up to a factor of 6.2), and metabolic cost rose across frequency by less than half as found experimentally (a factor of 1.42 vs. empirical 3.56). Current musculoskeletal models do not accurately predict the cost of cyclic reaching.

## Force-rate-dependent cost predicts point-to-point reaching motions and durations

We applied the energy cost from cyclic reaching to predict discrete, point-to-point reaching (*Figure 9*) of fixed and free durations. The prediction from trajectory optimization (*Equation 11*) was for a standard movement of fixed duration and distance (0.65 s and 30 cm, respectively; *Harris and Wolpert, 1998*), using the energy cost coefficients $c_W$ and $c_t$ derived from the primary experiment. This yielded bell-shaped velocities (*Figure 9*) similar to the minimum variance model and to empirical data (*Harris and Wolpert, 1998*). Also compared were minimum torque-rate (*Uno et al., 1989*), and minimum

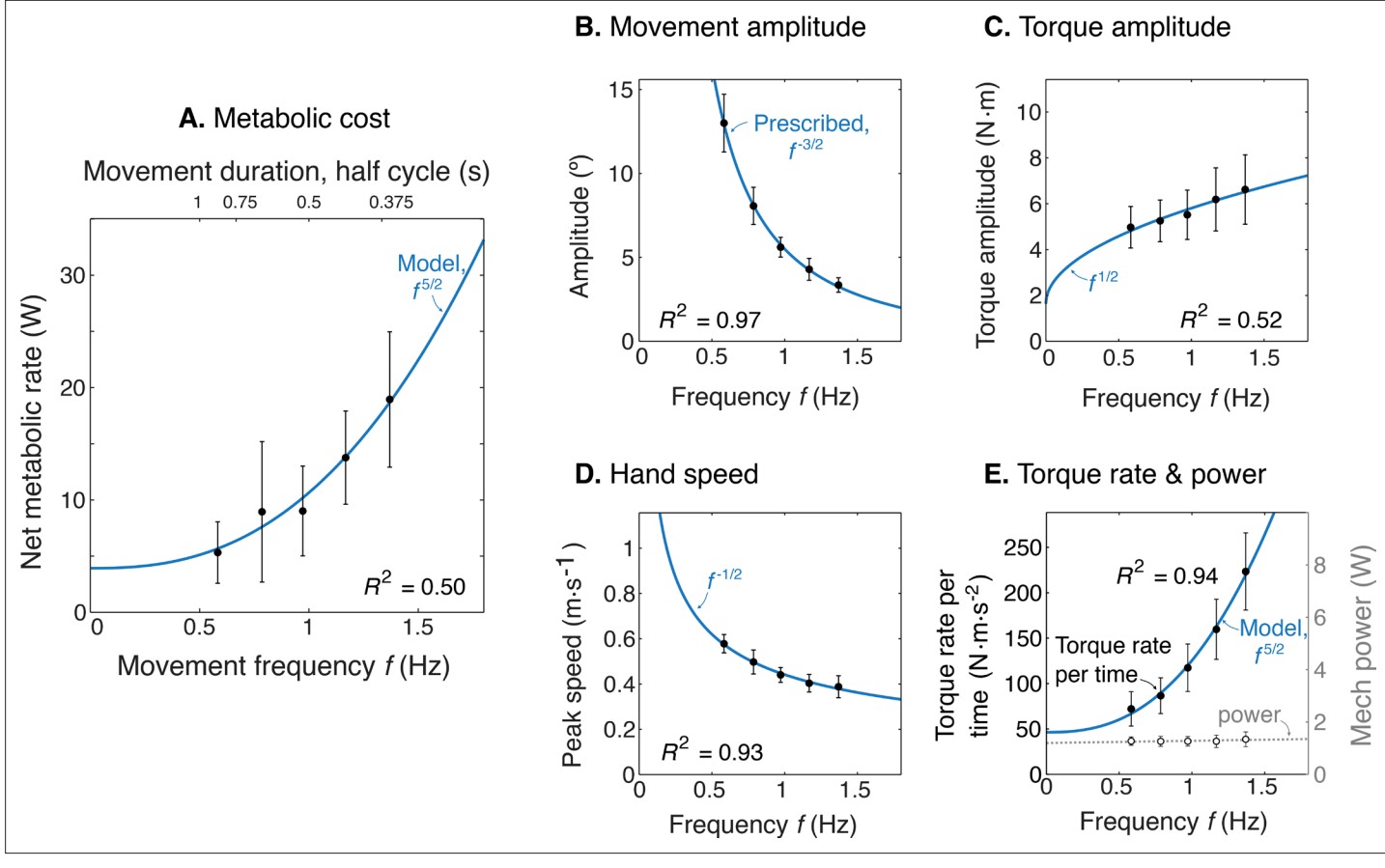

**Figure 5.** Experimental results as a function of movement frequency $f$. (**A**) Net metabolic power $\dot{E}$ vs. frequency $f$ (means ± s.d., $N = 10$), with predicted power law $f^{5/2}$ (solid line). (**B**) Movement amplitude and prescribed target $f^{-3/2}$. (**C**) Torque amplitude and prediction $f^{1/2}$. (**D**) Hand speed amplitude and prediction $f^{-3/2}$. (**E**) Amplitude of torque rate per time and prediction $f^{5/2}$, and mechanical power amplitude $\dot{W}$ and constant power prediction.

**Table 1.** Experimental results.
Linear mixed effects models were used to test model predictions from data. Listed for each quantity: predicted power law, estimated coefficient, 95% confidence interval (CI), $R^2$, and p-value.

| Quantity | Power law | Coefficient | 95% CI | $R^2$ | p | Intercept |
|---|---|---|---|---|---|---|
| Metabolic Power $\dot{E}$ (W) | $f^{5/2}$ | 6.72 | (4.58, 8.86) | 0.50 | 9.70E-9 | 3.93 |
| Movement amplitude $A$ (°) | $f^{3/2}$ | 5.97 | (5.66, 6.28) | 0.97 | 1.02E-39 | –0.47 |
| Peak speed amplitude (m.s$^{-1}$) | $f^{-1/2}$ | 0.43 | (0.39, 0.47) | 0.93 | 6.63E-29 | 0.01 |
| Torque amplitude (N.m) | $f^{1/2}$ | 8.34 | (5.77, 10.91) | 0.52 | 4.10E-9 | 1.63 |
| Positive mechanical power $\dot{W}$ (W) | $f^0$ | 1.20 | (0.85, 1.55) | | | |
| Torque rate per time $\dot{fT}$ (N.m.s$^{-2}$) | $f^{5/2}$ | 78.93 | (72.37, 85.48) | 0.94 | 2.19E-30 | 46.43 |
| EMG amplitude: Pec | $f^{3/2}$ | 0.17 | (0.12, 0.23) | 0.65 | 1.1E-6 | 0.17 |
| EMG amplitude: Delt | $f^{3/2}$ | 0.20 | (0.11, 0.27) | 0.56 | 1.5e-5 | 0.20 |

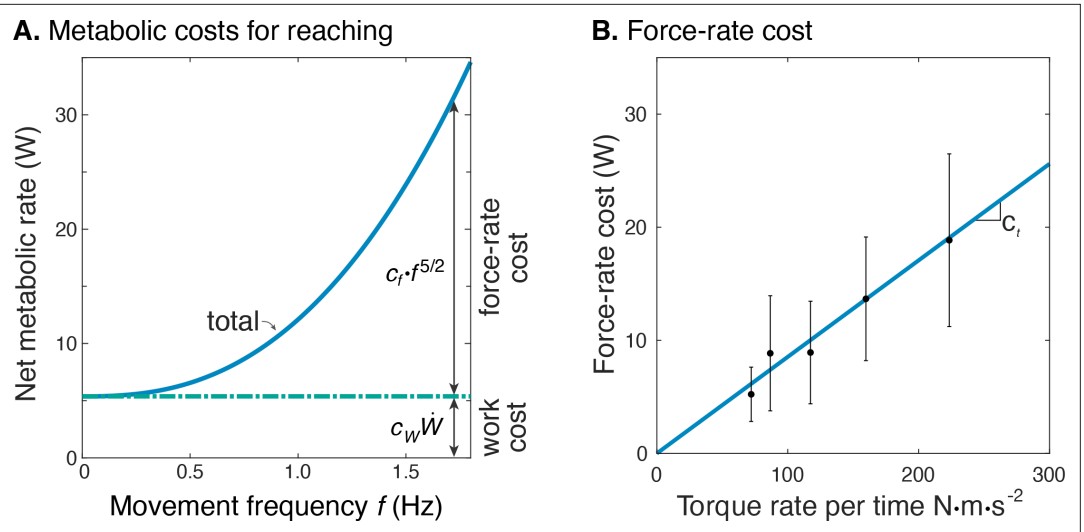

**Figure 6.** Estimated metabolic cost contributions from work and force-rate. (**A**) Net metabolic rate $\dot{E}$ vs. movement frequency $f$ for cyclic reaching, with contributions from force-rate cost ($c_f f^{5/2}$) and mechanical work ($c_W \dot{W}$). Coefficient $c_f$ was derived from experiment (**Figure 4**), whereas $c_W$ was specified as 4.2 to model a proportional cost for positive and negative mechanical work. (**B**) Force-rate cost (metabolic power $\dot{E}_{FR}$) is linearly related to amplitude of torque rate per time $f\dot{T}$, by coefficient $c_t$ determined from part A. and **Figure 4E**.

activation squared using a Hill-type muscle model (**Kistemaker et al., 2014**). Each objective approximately reproduces the empirical bell-shaped profile, with metabolic cost (**Equation 1**), torque-rate-squared, and activation-squared all having correlation coefficients above 0.8 (0.99, 0.98, and 0.82, respectively). The metabolic energy cost including empirically tested work and force-rate terms therefore predicts trajectories similar to other, non-energetic costs proposed previously, and to human data.

Similar predictions were made for different distances, leaving duration unconstrained (**Figure 10**). The predicted, optimal durations increased with movement distance, roughly similar to human preferred durations (**Reppert et al., 2018**). The associated trajectories also retained the bell-shaped velocities across all distances. The proposed metabolic energy cost, plus a penalty for long durations, therefore predicts both trajectories and durations roughly similar to human data.

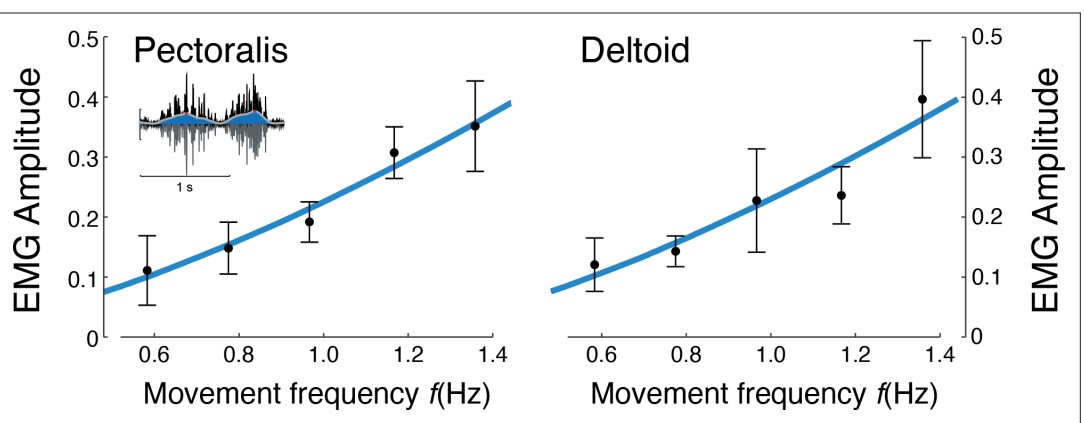

**Figure 7.** EMG amplitude vs.movement frequency $f$ during cyclic reaching. Inset figure depicts an example EMG rectified (black), filtered (blue), and amplitude (red). Pectoralis and deltoid EMGs (means ± s.d.; $N = 10$), with best-fit predictions curves (both $f^{3/2}$), $R^2 = 0.65$ and $R^2 = 0.56$, respectively.

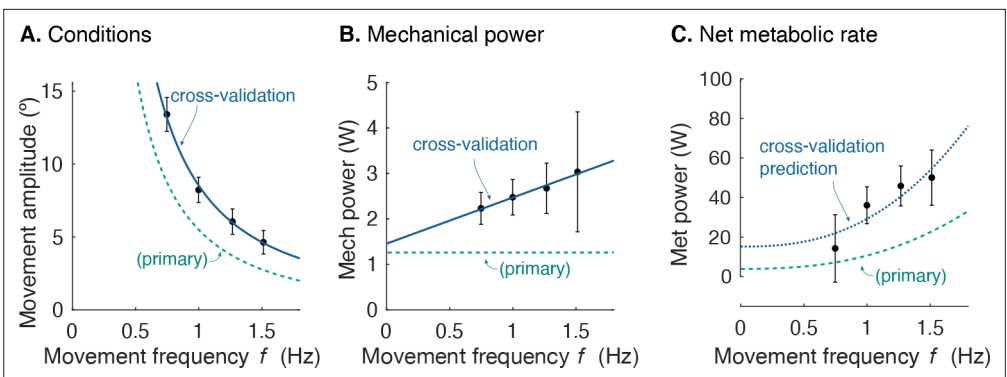

**Figure 8.** Cross-validation (CV) of force rate cost. (**A**) Amplitude of cyclic reaching condition (compared with primary experiment) vs. movement frequency $f$ and (**B**) positive mechanical power $\dot{W}$ vs. $f$. (**C**) Net metabolic rate $E$ vs. movement frequency $f$ for cross-validation (means ± s.d.; $N = 10$). Cross-validation conditions were such that average positive mechanical power $W$ increased slightly with $f$, unlike the primary experiment. Predicted metabolic rate for CV was determined using $c_t$ and $c_W$ from primary experiment (solid line).

## Discussion

We tested whether the metabolic cost of reaching movements is predicted by the hypothesized energetic cost including force-rate. Our experimental data showed a cost increasing with movement frequency as predicted with force-rate, more so than did the mechanical work performed. The same cost model was also cross-validated with a separate set of reaching movements, and predicts smooth reaching movements, similar to the minimum variance model. We interpret these findings as suggesting the force-rate hypothesis as an energetic basis for reaching movements.

The force-rate hypothesis explains the observed metabolic energy cost increases better than more conventionally recognized costs. For example, the cost of mechanical work alone cannot explain the higher cost at higher movement frequencies, because the rate of work remained fixed (**Figure 5**). A possible explanation is that the energetic cost per unit of work ($c_W$ in **Equation 1**) could increase with

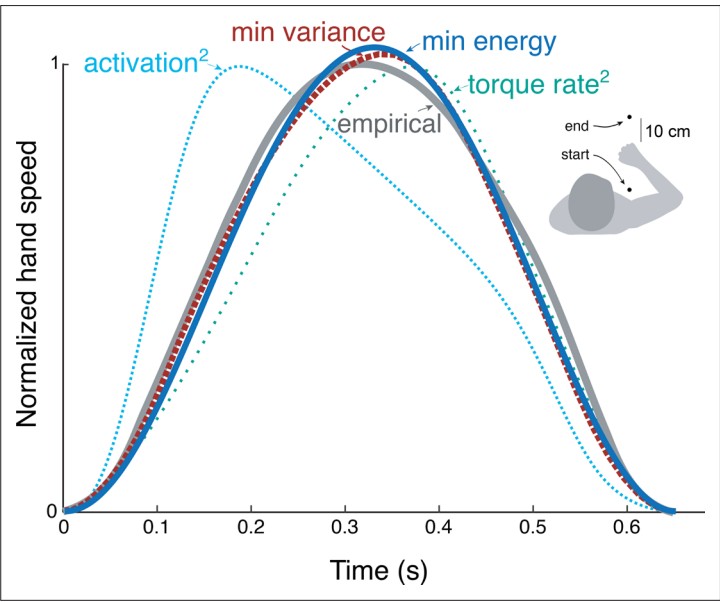

**Figure 9.** Hand speed trajectories vs.time for point-to-point movements predicted by various objective functions, compared to empirical, bell-shaped profiles. Minimization objectives include metabolic energy expenditure ('min energy' according to model proposed here), error variance (Harris & Wolpert), torque-rate squared (**Uno et al., 1989**, purple), and activation squared for a Hill-type muscle model. Minimum energy expenditure is the sum of work and force-rate costs (**Equation 1**), with coefficient $c_t$ identified from the primary experiment (**Figure 4**). All optima use the same initial and final targets and a fixed movement duration.

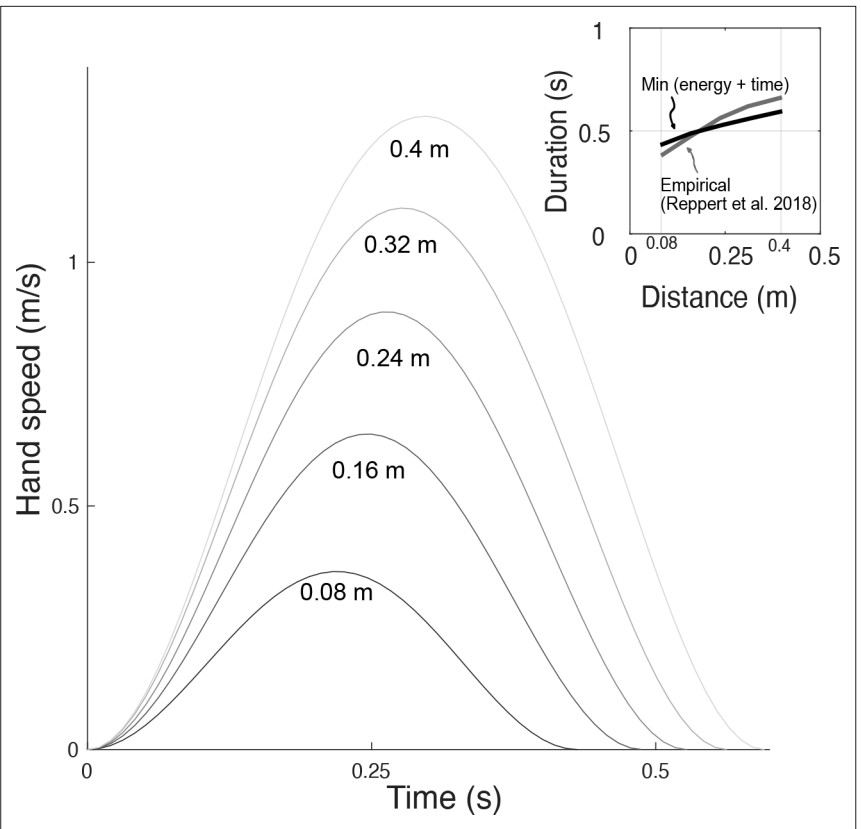

**Figure 10.** Hand speeds and movement time are predicted simultaneously by optimizing energetic cost and a (linear) cost of time. Hand speed as a function of time are plotted for five different reach distances. Inset: In recent empirical observations (adapted from *Reppert et al., 2018*, Figure 5), movement duration varies approximately linearly with reach distance.

faster movements, due to the muscle force-velocity relationship (*Barclay, 2015*). But the conditions here actually yielded slower hand speeds with higher frequencies (*Figure 5D*), and thus cannot explain the higher cost. Nor were our results explained by a current musculoskeletal model (*Umberger, 2010*), which drastically overestimated the overall cost and underestimated the increases with movement frequency. The proposed force-rate hypothesis thus explains these data better than previous quantitative models or relationships.

The force-rate hypothesis was also consistent with three other observations: (1) electromyography, (2) cross-validation, and (3) point-to-point reaching. First, muscle EMGs increased more rapidly (approximately with $f^{3/2}$ ; *Figure 7*) with movement frequency than did joint torques (approximately with $f^{1/2}$ ; *Figure 5C*). The proposed mechanism is that brief bursts of activation require greater active calcium transport (and thus greater energy cost), because muscle force production has slower dynamics than muscle activation (*van der Zee and Kuo, 2020*). Second, we cross-validated the primary experiment, by applying its cost coefficients ($c_t$ and $c_W$ , *Figure 6*) to predict an independent set of conditions. We found good agreement between cross-validation data and the force-rate prediction (*Figure 8*). The overall energy cost ($\dot{E}$ from *Equation 1*) depends on a particular combination of work, force, and movement frequency, yet only has one degree of freedom ($c_t$). Third, the force-rate hypothesis also explains discrete, point-to-point reaching. The characteristic bell-shaped velocity profile is predicted by optimal control, using the cost coefficients derived from cyclic movements (*Figure 9*). Moreover, movement duration is predicted to increase with distance, approximately similar to human reaches (*Reppert et al., 2018*). These observations serve as tests of the force-rate hypothesis, independently predicted by a single model.

The force-rate cost is surely not the sole explanation for reaching. The optimal control approach has been used to propose a variety of abstract mathematical objective functions that can predict movement. But there may be multiple objectives that predict similar behavior. As such, careful

experimentation (*Harris and Wolpert, 1998*; *Kawato, 1999*) was required to disambiguate minimum-variance from competing hypotheses such as minimum-jerk and -torque-change (*Kawato, 1999*). Similarly, the present study does not disambiguate force-rate from minimum-variance, since both predict similar point-to-point movements. In fact, minimum-variance also has some dependency on effort, albeit implicitly, due to the mechanism of signal-dependent noise (*Harris and Wolpert, 1998*). It also explains well the trade-off between movement speed and endpoint accuracy, where energy expenditure is unlikely to be important. However, the ambiguity also means that both variance and force-rate could potentially contribute to movement. It is quite possible that minimum variance dominates for fast and accurate movements, and energy cost for the trajectory and duration of slower ones, with both contributing to a unified objective for reaching.

Effort objectives have long been considered potential counterparts to the kinematic performance objective. For example, the integrated squared muscle force or activation or torque-change all emphasize effort and arm dynamics as explicit features for reaching (*Uno et al., 1989*). Effort is also important for selection of feedback control gains (*Kuo, 1995*; *Todorov and Jordan, 2002*), adaptation of coordination (*Emken et al., 2007*), identification of control objectives from data (*Vu et al., 2016*), and determination of movement duration (*Shadmehr et al., 2016*). The problem is that these manifestations of effort are abstract constructs with limited physiological basis, justified mainly by their ability to reproduce bell-shaped velocities through inverse optimization. However, multiple objectives can reproduce such velocities non-uniquely (*Figure 9*), making additional and independent tests important for disambiguating them. Accordingly, metabolic energy expenditure is a physiological, independently testable measure of effort.The change in metabolic cost during adaptation (*Huang et al., 2012*) and the effect of metabolic state on reaching patterns (*Taylor and Faisal, 2018*) strongly suggest a role for energy in reaching. The present study offers a means to incorporate a truly physiological effort cost into optimal control predictions for smooth and economical movements.

There is a measurable and non-trivial energetic cost for cyclic reaching. Even though the arms were supported by a planar manipulandum, at a movement frequency of 1.5 Hz, we observed a net metabolic rate of about 19 W. For comparison, the difference in cost between continuous standing and sitting is about 24 W (*Mansoubi et al., 2015*), making the reaching task nearly as costly as standing up. For each half-cycle reaching action, analogous to a point-to-point movement, the metabolic cost was about 3.5 J per arm. The cost may not be particularly high, but the nervous system may nonetheless prefer economical ways to accomplish a reaching task.

There are several limitations to this study. One is that energetic cost was experimentally measured for the whole body, and not distinguished at the level of the muscle. Force-rate was also estimated from joint torque and not from actual muscle forces. We therefore cannot eliminate other physiological processes as possible contributions to the observed energy cost. In addition, the hypothesized cost ($\dot{E}_{FR}$) is thus far a highly simplified, conceptual model for a muscle activation cost. More precise mechanistic predictions of this cost would be facilitated with specific models for muscle activation, myoplasmic calcium transport, and force delivery are needed (e.g. *Baylor and Hollingworth, 1998*; *Ma and Zahalak, 1991*). We also tested the force-rate cost in continuous reaches of fixed work, predominantly by the shoulder, and with the arms supported against gravity. Further studies are needed to test more ecological movements such as discrete reaching in arbitrary directions, and while interacting with objects.

The force-rate hypothesis suggests a substantial role for effort or energy expenditure in upper extremity reaching movements. Some form of effort cost is often employed to examine selection of feedback gains or muscle forces, and even generally expected for optimal control problems where maximal-effort actions are to be avoided (*Bryson and Ho, 1975*). And in the experimental realm, energy expenditure is regarded as a major factor in animal life and behavior (*Alexander, 1996*), even to the small scale of a single neural action potential (*Sterling and Laughlin, 2017*). Under the minimum-variance hypothesis alone, reaching seems unusually dominated by kinematics. But our results suggest that metabolic energy expenditure may have been over-shadowed by the minimum-variance hypothesis, because it makes similar predictions for point-to-point movements. There is need to both quantify and test the force-rate hypothesis more specifically, perhaps in combination with minimum-variance. Nonetheless, there is a meaningful energetic cost to reaching that can also explain the smoothness of reaching motions.

## Materials and methods

There were three main components to this study: (1) a simple cost model, (2) a set of human subjects experiments with cyclic reaching, and (3) an application of the model to predict discrete reaching trajectories. The model predicts that metabolic cost should increase with the hypothesized force-rate measure, particularly for faster frequencies of movement. Key to the experiment (*Figure 2*) was to isolate the hypothesized force-rate cost, by applying combinations of movement amplitude and frequency that control for the cost of mechanical work. This primary test was accompanied by a secondary, cross-validation test, with different combinations of movement amplitude and frequency. Finally, we applied this same force-rate cost to the prediction of discrete reaching movement trajectories. This was to test whether the energetic cost, derived from continuous, cyclic reaching movements, could also predict the smooth, discrete motions often found in the literature.

## Model predictions for force-rate hypothesis

We hypothesized that the energetic cost for reaching includes a cost for performing mechanical work, and another for the rate of force production. These costs are implemented on a simple, two-segment model of arm dynamics, actuated with joint torques. These torques perform work on the arm, at an approximately proportional energetic cost (*Margaria, 1976*) attributed to actin-myosin cross-bridge action (*Barclay, 2015*). The force-rate cost is hypothesized to result from rapid activation and deactivation of muscle, increasing with the amount of force and inversely with the time duration. It is attributable to active transport of myoplasmic calcium (*Bergström and Hultman, 1988*; *Hogan et al., 1998*), where more calcium is required for higher forces and/or shorter time durations, hence force-rate (*Doke and Kuo, 2007*).

For the simple motion employed here, the prediction of the total metabolic energy $E$ consumed per movement is the sum of costs for work and force-rate,

$$E = c_W W + E_{FR} \tag{1}$$

where $W$ is the positive mechanical work per movement, $c_W$ the metabolic cost per unit of work, and $E_{FR}$ is the hypothesized force-rate cost

$$E_{FR} = c_f \dot{F} \tag{2}$$

where $\dot{F}$ denotes the amplitude of force-rate (time-derivative of muscle force) per movement, and $c_f$ is the energetic cost for force-rate. This cost is to be distinguished from the earlier torque-change hypothesis (*Uno et al., 1989*), which integrates a sum-squared force-rate over time, and which had no hypothesized relationship to metabolic energetic cost. During cyclic reaching, the peak force-rate $\dot{F}$ increases with both force amplitude and the frequency of cyclic movement. Here, positive and negative work are performed in equal magnitudes, and so their respective costs are lumped together into a single proportionality $c_W$. We assigned $c_W$ a value of 4.2, from empirical mechanical work efficiencies of 25% for positive work and –120% for negative work (*Margaria et al., 1963*).

The work and force of the cyclic reaching movements about the shoulder are predicted by a simple model of arm dynamics. In the horizontal plane of a manipulandum supporting the arm,

$$T(t) = I\ddot{\theta}(t) \tag{3}$$

with shoulder angle $\theta(t)$, shoulder torque $T(t)$ (treated as proportional to muscle force), and rotational inertia $I$. Applying sinusoidal motion at amplitude $A$ and movement frequency $f$ (in cycle/s),

$$\theta(t) = A cos 2\pi f t. \tag{4}$$

The torque is therefore

$$T(t) = -4\pi^2 I A f^2 cos 2\pi f t \tag{5}$$

and amplitude of mechanical power $\dot{W}$

$$\dot{W} \propto A^2 f^3 \tag{6}$$

We apply a particular movement condition, termed the *fixed power* constraint (*Figure 2A*), where the average positive mechanical power is kept fixed across movement frequencies, so that the hypothesized force-rate cost will dominate energetic cost (*Figure 3A*). This is achieved by constraining amplitude to decrease with movement frequency (*Figure 3B*),

$$A \propto f^{\frac{-3}{2}}$$ (7)

This fixed power condition also means that hand (endpoint) speed, proportional to $\dot{\theta}$ , should have amplitude varying with $f^{-1/2}$ , and torque amplitude with $f^{1/2}$ (*Figure 3C and D*).

Applying fixed power to the force-rate cost yields the energetic cost prediction. Torque-rate amplitude $\dot{T}$ with *Equation 2* and *Equation 7* yields

$$\dot{T} = b \cdot f^{\frac{3}{2}}$$ (8)

where $b$ is a constant coefficient. The proportional cost per movement is therefore (*Equation 2*)

$$E_{FR} = c_f \cdot f^{\frac{3}{2}}$$ (9)

where $c_f$ is a constant coefficient across conditions. Experimentally, it is most practical to measure metabolic power $\dot{E}$ (*Figure 3a*) in steady state. Multiplying $E$ (cost per movement, *Equation 2*) by $f$ (movement cycles per time) yields the predicted proportionality for average metabolic power,

$$\dot{E}_{FR} = c_f \cdot f^{5/2}.$$ (10)

The net metabolic rate $\dot{E}$ is expected to increase similarly, but with an additional offset for the constant work cost $\dot{E}_W$ under the fixed-power constraint (*Figure 3A*). Finally, the metabolic energy per time associated with force-rate would be expected to increase directly with torque-rate per time $f \cdot \dot{T}$ ,

$$\dot{E}_{FR} = c_t \cdot f \cdot \dot{T}$$ (11)

where movement frequency $f$ represents cycles per time, and coefficient $c_t$ is equal to $c_f$ divided by $b$.

This force-rate coefficient is not specific to cyclic movements alone. The general metabolic cost model (*Equations 1; 2*) is potentially applicable to point-to-point and other motions, with different amounts of work and force-rate. The force-rate cost $\dot{E}_{FR}$ is independent of mechanical work, and may be predicted using the cost coefficient derived from cyclic experiments. The model may therefore make testable predictions of energetic cost even for movements that are acyclic and not constrained to fixed power.

## Experiments

We measured the metabolic power expended by healthy adults ($N$ = 10) performing cyclic movements at a range of speeds but fixed power (*Equation 7*). We tested whether metabolic power would increase with the hypothesized force-rate cost $\dot{E}_{FR}$, in amount not explained by mechanical work. We also characterized the mechanics of the task in terms of kinematics, shoulder torque amplitude, and force-rate for shoulder muscles. These were used to test whether the mechanics were consistent with the model of arm dynamics, and whether force-rate increased as predicted (*Equations 7–10*). We first describe a primary experiment with fixed power conditions, followed by an additional cross-validation experiment. All subjects provided written informed consent, as approved by University of Calgary Ethics board.

Subjects performed cyclic bimanual reaching movements in the horizontal plane, with the arms supported by a robotic exoskeleton (KINARM, BKIN Technologies, Inc). The movements were cyclic and bimanually symmetrical to induce steady energy expenditure sufficient to be distinguished easily by expired gas respirometry. The exoskeleton was used to counteract gravity in a low-friction environment (with no actuator loads), and to measure kinematics, from which shoulder and elbow joint torques were estimated using inverse dynamics. Subjects were asked to move each arm between a pair of targets, reachable mostly by medio-lateral shoulder motion, with relatively little elbow motion (less than 1 deg average excursion across all conditions). A single visual cursor (5 mm in diameter) was displayed for the right hand, along with one pair of visual targets (circles 2.5 cm in diameter), all optically projected onto the movement plane. To encourage equal bimanual motion, the cursor's position

was not for one hand alone, but rather computed as an average of right and left arm joint angles, making it insufficient to move one arm alone.

Timing was set with a metronome beat for reaching each of the two targets, and amplitude by adjusting the distance between the targets. Prior to data collection, subjects completed a 20-min familiarization session (up to 48 hr before the experiment) where they received task instructions and briefly practiced each of the tasks.

The primary experiment was to test for the predicted energetic cost for reaching, in five conditions of cyclic reaching at increasing frequency and decreasing amplitude. The frequencies were 0.58, 0.78, 0.97, 1.17, 1.36 Hz, and amplitudes were 12.5, 8, 5.8, 4.4, 3.5°, respectively. These cyclic movements were chosen to be of moderate hand speed, with peak speeds between 0.4 and 0.6 m/s.

We estimated metabolic rate using expired gas respirometry. Subjects performed each condition for 6 min, analyzing only the final 3 min of data for steady-state aerobic conditions, with standard equations used to convert O2 and CO2 rates into metabolic power (**Brockway, 1987**).We report net metabolic rate $\dot{E}$ for bimanual movement, defined as gross rate minus the cost of quiet sitting (obtained in a separate trial, 98.6 ± 11.5 W, mean ± s.d.).

We also recorded arm segment positions and electromyographs simultaneously at 1000 Hz. These included kinematics from the robot, and electromyographs (EMGs) from four muscles (pectoralis lateral, posterior deltoid, biceps, triceps) in a subset of our subjects (five subjects in primary experiment, five in cross-validation). The EMGs were used to characterize muscle activation and co-activation.

The metabolic cost hypothesis was tested using a linear mixed-effects model of net metabolic power. This included the hypothesized force-rate cost (**Equation 10**) as a fixed effect, yielding coefficient $c_f$ for the force-rate term proportional to $f^{5/2}$ . A constant offset was included for each subject as a random effect. In addition, the force-rate cost $\dot{E}_{FR}$ was estimated by subtracting the fixed mechanical work cost $\dot{E}_W$ from net metabolic power $\dot{E}$ , and then compared against torque rate amplitude per time (**Equation 11**). Sample size was appropriate to yield a statistical power of 0.99 based on statistical characteristics of previous reaching studies of metabolic cost (**Wong et al., 2018**). Both the main experiment and cross validation experiment were performed a single time.

We tested expectations for movement amplitude and other quantities from kinematic data. Hand velocity was filtered using a bi-directional lowpass Butterworth filter (first order, 12 Hz cutoff). Shoulder and elbow torques were computed using inverse dynamics, based on KINARM dynamics (BKIN Technologies, Kingston), and subject-specific inertial parameters (**Winter, 1990**). The approximate rotational inertia of a single human arm and exoskeleton about the shoulder was estimated as 0.9 kg·m$^2$. The positive portion of bimanual mechanical power was integrated over total movement duration and divided by cycle time, yielding average positive mechanical power. Linear mixed-effects models were used to characterize the power-law relations for mechanical power, movement amplitude, movement speed, torque amplitude, and torque rate amplitude (**Figure 3**). The latter was estimated by integrating the torque rate amplitude per time (**Equation 11**) for each joint, and then summing the two. The force-rate hypothesis was also tested by comparing $\dot{E}_{FR}$ with torque rate per time (**Figure 3A**), assuming torque is proportional to muscle force.

Electromyographs were used to test for changes in muscle activation and co-activation. Data were mean-centred, low-pass filtered (bidirectional, second order, 30 Hz cutoff), rectified, and low-pass filtered again (**Roberts and Gabaldón, 2008**), from which the EMG amplitude was measured at peak and then normalized to each subject's maximum EMG across the five conditions. We expected EMG amplitude to increase with muscle activation, with simplified first-order dynamics between activation (EMG) and muscle force production (**van der Zee and Kuo, 2020**). This treats the rate-limiting step of force production as a low-pass filter, so that greater activation or EMG amplitudes would be needed to produce a given force at higher waveform frequencies. The first-order dynamics mean that EMG would be expected to increase with torque rate $f^{3/2}$ rather than torque, as tested with a linear mixed-effects model. We also computed a co-contraction index for EMG, in which the smallest value of antagonist muscle pairs was computed over time, and then integrated for comparison across conditions (**Gribble et al., 2003**). All statistical tests were performed with threshold for significance of $p < 0.05$.

As a cross-validation test of the force-rate cost, we tested the generalizability of coefficient $c_t$ against a second set of conditions with a separate set of subjects (also $N$ = 10; two subjects participated in both sets). The conditions were slightly different: frequencies ranging 0.67–1.3 Hz and

amplitudes 12.5–4.42°, which resulted in higher mechanical work and force-rate. We applied the model (*Equation 1*, *Equation 11*) and $c_t$ coefficient identified from the primary experiment to the cross-validation conditions. As a further test of the central hypothesis, we expected the model to roughly predict trends regarding mechanical and metabolic rates for the cross-validation conditions.

### Estimation of elastic energy storage in shoulder muscles

We estimated the resonant frequency of cyclic reaching, to account for possible series elasticity in shoulder actuation. Series elasticity could potentially store and return energy and thus require less mechanical work from muscle fascicles. We estimated this contribution from resonant frequency, obtained by asking subjects to swing their arms back and forth rapidly at large amplitudes (at least 15°) for 20 s, and determining the frequency of peak power (PWelch in Matlab). We then used this to estimate torsional series elasticity, and the passive contribution to mechanical power.

### Musculoskeletal model to simulate experimental conditions

We tested whether a Hill-type musculoskeletal model could explain the metabolic cost of cyclic reaching. The hypothesized force-rate is not explicitly included in current models of energy expenditure, and would not be expected to explain the experimental metabolic cost. We therefore tested an energetics model available in the OpenSim modeling system (*Seth et al., 2018*; *Uchida et al., 2016*; *Umberger, 2010*), applied to a model of arm dynamics with six muscles (*Kistemaker et al., 2014*). We used trajectory optimization to determine muscle states and stimulations, with torques from inverse dynamics as a tracking reference. Optimization was performed using TOMLAB/SNOPT (Tomlab Software AB, Sweden; *Gill et al., 2002*), to minimize mean-squared torque error, squared stimulation level, and squared stimulation rate. The optimized muscle states were then fed into the metabolic cost model (*Umberger, 2010*).

### Model of point-to-point reaching movements

The force-rate cost hypothesis was also used to predict point-to-point reaching movements and their durations. Here, we form an overall objective function $J$ that includes the energetic cost per movement $E$ (*Equation 1*) as a physiological effort term, and a simple penalty proportional to movement duration $t_f$ . This may be regarded as an adaptation of Shamehr et al.'s (2016) hypothesis that duration is a trade-off between effort and (the inverse of) a temporally discounted reward. The overall objective is thus:

$$J\left(\theta\left(t\right),\tau\left(t\right)\right) = E\left(\theta\left(t\right),\tau\left(t\right)\right) + k \cdot t_f \tag{12}$$

where the energy expenditure is expressed a function of joint angle $\theta\left(t\right)$ and torque $\tau\left(t\right)$ trajectories, and duration is penalized with proportionality $k$. Minimization of this objective can predict point-to-point reaching trajectories both of fixed duration (by constraining $t_f$) and of free duration. We show that this objective predicts smooth, bell-shaped velocity profiles similar to minimum-variance, as well as durations increasing with movement distance.

We used this objective in trajectory optimization of planar, two-joint reaching movements. For fixed duration $t_f$ , the objective $J$ depends only on energetic cost $E$, with the mechanical work and force-rate terms expressed as a time integral for both joints:

$$E\left(\theta\left(t\right),\tau\left(t\right)\right) = \sum_{i=1}^{2} \int_0^{t_f} \left( \left| \dot{\theta}_i \tau_i \right|_+ + \left| c_t \ddot{\tau}_i \right|_+ \right) dt \tag{13}$$

with joint torques $\tau_i$ ($i = 1, 2$ for elbow and shoulder, respectively). To compare with the minimum variance model of *Harris and Wolpert, 1998*, we used a similar straight reaching movement of amplitude 30 cm and duration $t_f$ of 650 ms. We also used the empirically estimated $c_t$ from the cyclic reaching experiment (assuming the same coefficient for both shoulder and elbow), along with a point-to-point constraint to have zero initial and final acceleration of the hand, again using TOMLAB. The predicted hand velocity trajectory was qualitatively compared with the empirical bell-shaped velocity from minimum variance (*Harris and Wolpert, 1998*).

We also examined movements of unconstrained duration, which have been shown to take longer with greater movement distance (*Reppert et al., 2018*). We selected $k$ so that the average movement speed was approximately equal to the average preferred movement duration across the empirically

measured reach speeds ($k$ = 25 J/s). We qualitatively compared the trajectories and durations from model against data for movements ranging 8–40 cm (*Reppert et al., 2018*).

## Acknowledgements

This work was funded by NSERC (Discovery and CRC Tier 1), Dr. Benno Nigg Research Chair, and Alberta Health Trust. We acknowledge Dinant Kistemaker for sharing simulation code for Hill-type muscle model energetics.

## Additional information

### Funding

| Funder | Grant reference number | Author |
|---|---|---|
| University of Calgary | Benno Nigg Chair | Arthur D Kuo |
| Natural Sciences and Engineering Research Council of Canada | | Arthur D Kuo |
| Alberta Health Services | | Arthur D Kuo |
| Natural Sciences and Engineering Research Council of Canada | | Tyler Cluff |

The funders had no role in study design, data collection and interpretation, or the decision to submit the work for publication.

### Author contributions

Jeremy D Wong, Conceptualization, Data curation, Formal analysis, Investigation, Methodology, Project administration, Resources, Software, Visualization, Writing - original draft, Writing – review and editing; Tyler Cluff, Formal analysis, Funding acquisition, Methodology, Writing – review and editing; Arthur D Kuo, Conceptualization, Funding acquisition, Project administration, Supervision, Visualization, Writing - original draft, Writing – review and editing

### Author ORCIDs

Jeremy D Wong http://orcid.org/0000-0001-5564-1794

### Ethics

Human subjects: Informed consent was obtained from all subjects and the Health Research Ethics Board approved of all procedures (REB18-1521).

### Decision letter and Author response

Decision letter https://doi.org/10.7554/eLife.68013.sa1
Author response https://doi.org/10.7554/eLife.68013.sa2

## Additional files

### Supplementary files

• Transparent reporting form

### Data availability

Data has been deposited to Dryad Digital Repository, accessible here: doi:http://doi.org/10.5061/dryad.qfttdz0gn.

The following dataset was generated:

| Author(s) | Year | Dataset title | Dataset URL | Database and Identifier |
|---|---|---|---|---|
| Wong JD, Cluff T, Kuo AD | 2021 | The energetic basis for smooth human arm movements | http://dx.doi.org/10.5061/dryad.qfttdz0gn | Dryad Digital Repository, 10.5061/dryad.qfttdz0gn |

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
