## [Editor Report]

This paper will be of interest to researchers in the fields of biomechanics, movement control, and decision making. A novel, mechanistic model of metabolic cost is presented to account for a phenomenon not explained by current models of metabolic energy. This is followed by a demonstration of how this metabolic model can improve our understanding of movement control by revealing an energetic basis for smooth movements.

---

## [Decision Letter]

**Decision letter after peer review:**

Thank you for submitting your article "The energetic basis for smooth human arm movements" for consideration by *eLife*. Your article has been reviewed by 2 peer reviewers, one of whom is a member of our Board of Reviewing Editors, and the evaluation has been overseen by Richard Ivry as the Senior Editor. The following individual involved in review of your submission has agreed to reveal their identity: Mazen Al Borno (Reviewer #2).

Essential revisions:

1) The authors claim to present a theory of movement control that unifies smoothness and duration. They currently only show that their metabolic cost model can explain smoothness. However, their metabolic cost model on its own cannot explain the speed-accuracy tradeoff and minimum variance on its own cannot explain duration. Thus to support a unifying theory, they need to show that their metabolic cost model and minimum variance combined can explain all three movement properties: smoothness, the speed-accuracy tradeoff and duration. They also need to show that this new unifying model performs better than models using a combination of minimum variance and other effort models (such as mech work, hill muscle models, and torque-change). Alternatively, instead of this additional analysis, the authors could temper their claims of a unifying theory and instead focus on their result that smoothness can also be explained by energy minimization. Although a weaker finding, this is also a an important contribution to the field.

2) A comparison of the predicted velocity profiles by their metabolic model and the predictions of models based on squared muscle activations, torque rate, and Hill-type muscle model is needed.

3) To support their claim that energy explains duration, the authors should demonstrate the ability of their metabolic model to explain movement duration, and compare its predictions to the predictions of other effort models (mechanical work, squared muscle activations, torque rate, Hill-type muscle models). This analysis should also include whether their metabolic model predicts a non-zero metabolically optimal movement duration and if so, how the metabolically optimal duration scales with movement distance.

*Reviewer #1 (Recommendations for the authors):*

1. For the predictions of the bell-shaped velocity profile, it is not clear how the effort cost was represented in the cost function. Equation (1) represents the effort cost per movement, and even Equation (9) seems to represent the average energy per time associated with force-rate, since it is a function of the frequency of the contraction and the amplitude of the torque rate of the contraction. Thus, there is no expression for effort cost as a function of instantaneous time (i.e. similar to effort models based on torque rate squared or force squared or activation squared), rather it is calculated based on the total mechanical work and the frequency (or duration) of the movement. It would help to have the authors clarify how effort was represented in their optimization approach.

2. Additional figures should be provided plotting individual and average subject trajectory data and emg data across the different conditions. A figure of the experimental setup and visual feedback should also be provided. It is unclear how a single cursor was used to show the average joint angles.

3. As written, it seems that Equations 7 and 9 are dependent upon the fixed power assumption. This would mean that the fitted coefficient in Equation 9 as a whole could only apply when comparing conditions with identical power. Specifically, Equation (2) states the hypothesis that the force-rate per contraction increases proportionally with the amplitude of force rate (or amplitude of torque rate). For the model in Equation (3) and the sinusoidal motion shown, the amplitude of torque rate increases with Af3 (the product of the amplitude of the motion A, and the frequency, f, cubed). In the fixed power condition, the amplitude of the motion is a function of the frequency, so torque rate increases proportionally with bf3/2, where b is a fitted coefficient. Thus, Equation 7 (and Equation 9) which related amplitude of torque rate to cost per contraction, and cost per contraction per unit time, appear specific to the fixed power constraint. The authors should more clearly explain how this derivation, and the fitted coefficient, can generalize to non-fixed power conditions. I assume that any differences in mechanical power are absorbed in the first term in Equation (1), thus the coefficient would generalize, but this should be clarified.

4. Related to the point above, more effort should be made to explain these frequency-based experiments and findings to a single movement with varying distance and duration.

5. The authors should also plot metabolic cost vs movement duration for the different distances, as this is more of a standard form. This would allow for an easier understanding of the data in the context of a single reach.

6. Related to the point above, the authors do not address how their metabolic model predictions compare to published data on the metabolic cost of a single reach as a function of distance and duration. It would strengthen the impact of their findings to demonstrate whether the current dataset exhibits similar relationships between metabolic cost, movement duration and distance.

7. The authors seem to be underselling the cross-validation. If we understand correctly, the coefficient fitted in one experiment with one group of subjects, was able to explain data in a different experiment with different subjects with different frequencies and amplitudes, and the conditions had different power requirements. This is a lovely finding and should be highlighted more strongly. To help highlight the strengths of the cross-validation, there should be a more detailed explanation of the purpose of the cross-validation: to check whether the coefficient generalizes to different subjects as well as different frequencies/amplitudes/mech.power conditions.

8. A more quantitative comparison is needed (in addition to the figure presented in figure 8), when comparing the metabolic model velocity profile predictions to empirical data and the minimum variance predictions.

9. Why was the experiment a bilateral movement? Did the authors divide the measured power by 2? I couldn't find mention of this in the methods.

10. It seems that the elbow is externally fixed during the movement. Otherwise the subjects would need to stabilize the elbow and the model used of a single joint/rigid body would be inappropriate. However this is not clear from the text. Please confirm.

11. Sometimes the authors refer to 'energy cost per contraction' and sometimes 'energy cost per movement'. Please clarify the distinction, if any.

*Reviewer #2 (Recommendations for the authors):*

The paper states that minimizing squared muscle activations does not predict smooth movements. My understanding is that optimal feedback theory does predict smooth movements (Todorov 2002; Al Borno 2020). For instance, in your model, if you included only the squared muscle activations term (or just used the Hill-type model), would you obtain a non-smooth bell-shaped velocity profile?

As pointed out in the paper, in minimum-variance theory, an effort term is implicitly there. This could indicate that minimum-variance theory is sufficient to explain all the data (in addition to the speed-accuracy tradeoff, which this work does not address). If this is incorrect, it would be useful for the authors to point out where minimum-variance theory falls short. For instance, recent work by Nagamori et al., 2021 questions the main assumption behind the theory.

---

## [Author Response]

Essential revisions:1) The authors claim to present a theory of movement control that unifies smoothness and duration. They currently only show that their metabolic cost model can explain smoothness. However, their metabolic cost model on its own cannot explain the speed-accuracy tradeoff and minimum variance on its own cannot explain duration. Thus to support a unifying theory, they need to show that their metabolic cost model and minimum variance combined can explain all three movement properties: smoothness, the speed-accuracy tradeoff and duration. They also need to show that this new unifying model performs better than models using a combination of minimum variance and other effort models (such as mech work, hill muscle models, and torque-change). Alternatively, instead of this additional analysis, the authors could temper their claims of a unifying theory and instead focus on their result that smoothness can also be explained by energy minimization. Although a weaker finding, this is also a an important contribution to the field.

We were unclear about the unified objective, and have amended the discussion to make it clearer that energy economy is not treated as the sole objective for reaching movements. For example:

“…both variance and force-rate could potentially contribute to movement. It is quite possible that minimum variance dominates for fast and accurate movements, and energy cost for the trajectory and duration of slower ones, with both contributing to a unified objective for reaching.”

We have also added more explanation for how economy can predict duration (Figure 10), and have added discussion about how “metabolic energy expenditure is a physiological, independently testable measure of effort” (in contrast to more abstract objectives such as minimum activation or torque change).

2) A comparison of the predicted velocity profiles by their metabolic model and the predictions of models based on squared muscle activations, torque rate, and Hill-type muscle model is needed.

As suggested, we have included these predictions in an updated figure (now Figure 9). We also expanded discussion regarding them: “…multiple objectives can reproduce such velocities non-uniquely, making additional tests important for disambiguating them. The advantage of metabolic energy expenditure is that it is a physiological, independently testable measure of effort.

3) To support their claim that energy explains duration, the authors should demonstrate the ability of their metabolic model to explain movement duration, and compare its predictions to the predictions of other effort models (mechanical work, squared muscle activations, torque rate, Hill-type muscle models). This analysis should also include whether their metabolic model predicts a non-zero metabolically optimal movement duration and if so, how the metabolically optimal duration scales with movement distance.

We have added simulations using force-rate cost to predict preferred movement duration as a function of distance (new Figure 10), along with comparison to recent experimental literature (Reppert et al., 2018). The figure shows how predicted duration is non-zero and increases with distance. We did not test the other objectives, because to our knowledge they have not been proposed to predict duration.

Reviewer #1 (Recommendations for the authors):1. For the predictions of the bell-shaped velocity profile, it is not clear how the effort cost was represented in the cost function. Equation (1) represents the effort cost per movement, and even Equation (9) seems to represent the average energy per time associated with force-rate, since it is a function of the frequency of the contraction and the amplitude of the torque rate of the contraction. Thus, there is no expression for effort cost as a function of instantaneous time (i.e. similar to effort models based on torque rate squared or force squared or activation squared), rather it is calculated based on the total mechanical work and the frequency (or duration) of the movement. It would help to have the authors clarify how effort was represented in their optimization approach.

We agree, and have added the explicit energy equation that is minimized during the fixedtime optimization, on page 16.

2. Additional figures should be provided plotting individual and average subject trajectory data and emg data across the different conditions. A figure of the experimental setup and visual feedback should also be provided. It is unclear how a single cursor was used to show the average joint angles.

We have added text to explain the motion of the cursor located in the methods section Page 13. We neglected to note clearly that the second panel includes individual subject data, and we have amended the caption appropriately. Finally, we have added an additional figure (new figure 4) of average reaches.

3. As written, it seems that Equations 7 and 9 are dependent upon the fixed power assumption. This would mean that the fitted coefficient in Equation 9 as a whole could only apply when comparing conditions with identical power. Specifically, Equation (2) states the hypothesis that the force-rate per contraction increases proportionally with the amplitude of force rate (or amplitude of torque rate). For the model in Equation (3) and the sinusoidal motion shown, the amplitude of torque rate increases with Af3 (the product of the amplitude of the motion A, and the frequency, f, cubed). In the fixed power condition, the amplitude of the motion is a function of the frequency, so torque rate increases proportionally with bf3/2, where b is a fitted coefficient. Thus, Equation 7 (and Equation 9) which related amplitude of torque rate to cost per contraction, and cost per contraction per unit time, appear specific to the fixed power constraint. The authors should more clearly explain how this derivation, and the fitted coefficient, can generalize to non-fixed power conditions. I assume that any differences in mechanical power are absorbed in the first term in Equation (1), thus the coefficient would generalize, but this should be clarified.

We have added specific equations to show how Equation (2) applies to point-to-point movements (new Equations 10 and 11), along with clarification how the coefficient to force-rate is independent of the fixed power constraint (new subsection Model of point-to-point reaching). In Methods:

“This force-rate coefficient is not specific to cyclic movements alone… The model may therefore make testable predictions of energetic cost even for movements that are acyclic and not constrained to fixed power.”

4. Related to the point above, more effort should be made to explain these frequency-based experiments and findings to a single movement with varying distance and duration.

The new subsection Model of point-to-point reaching (page 16) provides greater detail on single movements. Briefly, we applied the coefficient found from cyclic movements to predict discrete movements, which use a state-dependent form of the objective, plus different boundary constraints at movement start and end.

5. The authors should also plot metabolic cost vs movement duration for the different distances, as this is more of a standard form. This would allow for an easier understanding of the data in the context of a single reach.

We have added a second axis to our (new Figure 5)A to include movement amplitude.

6. Related to the point above, the authors do not address how their metabolic model predictions compare to published data on the metabolic cost of a single reach as a function of distance and duration. It would strengthen the impact of their findings to demonstrate whether the current dataset exhibits similar relationships between metabolic cost, movement duration and distance.

The new point-to-point subsection adds detail about duration. We have included predictions of increasing movement time with distance (new Figure 10), which re-uses the force-rate coefficient found in experiment.

7. The authors seem to be underselling the cross-validation. If we understand correctly, the coefficient fitted in one experiment with one group of subjects, was able to explain data in a different experiment with different subjects with different frequencies and amplitudes, and the conditions had different power requirements. This is a lovely finding and should be highlighted more strongly. To help highlight the strengths of the cross-validation, there should be a more detailed explanation of the purpose of the cross-validation: to check whether the coefficient generalizes to different subjects as well as different frequencies/amplitudes/mech.power conditions.

Yes, that understanding is correct. We have modified the Methods topic sentence:

“As a cross-validation test of the force-rate cost, we tested the generalizability of coefficient ;;_!_ against a second set of conditions with a separate set of subjects”.

Later in same paragraph:

“we expected the model to roughly predict trends regarding mechanical and metabolic rates for the cross-validation conditions.”

8. A more quantitative comparison is needed (in addition to the figure presented in figure 8), when comparing the metabolic model velocity profile predictions to empirical data and the minimum variance predictions.

We have added a quantitative comparison of the different models to capture bell-shapes, page 14. We report correlation coefficients for the models. Even though our model yielded the highest correlation (0.99), we state in Discussion:

“However, multiple objectives can reproduce such [bell-shaped] velocities non-uniquely (page 10)… metabolic energy expenditure is a physiological, independently testable measure of effort.”

9. Why was the experiment a bilateral movement? Did the authors divide the measured power by 2? I couldn't find mention of this in the methods.

In Methods, we have added:

**“**The movements were cyclic and bimanually symmetrical to induce steady energy expenditure sufficient to be distinguished easily by expired gas respirometry.”

Also in Methods:

“We report net metabolic rate ;;; for bimanual movement…” We did not divide by 2, except in Discussion, where we have added “And per reaching movement, the metabolic cost (at two movements per cycle) was about 3.5 J per arm.”

10. It seems that the elbow is externally fixed during the movement. Otherwise the subjects would need to stabilize the elbow and the model used of a single joint/rigid body would be inappropriate. However this is not clear from the text. Please confirm.

We have amended the text to clarify that both shoulder and elbow torques were computed from inverse dynamics, and the same force-rate coefficient identified for both. That same coefficient was then used to predict point-to-point movements.

11. Sometimes the authors refer to 'energy cost per contraction' and sometimes 'energy cost per movement'. Please clarify the distinction, if any.

We have simplified to “cost per movement,” and in Discussion consider the cost for each half-cycle movement.

Reviewer #2 (Recommendations for the authors):The paper states that minimizing squared muscle activations does not predict smooth movements. My understanding is that optimal feedback theory does predict smooth movements (Todorov 2002; Al Borno 2020). For instance, in your model, if you included only the squared muscle activations term (or just used the Hill-type model), would you obtain a non-smooth bell-shaped velocity profile?

We added now Figure 9 includes a comparison to hill-model activation-squared and other objectives. We agree that some specific models can produce bell-shaped velocities, and have amended the text accordingly. We also note:

“However, multiple objectives can reproduce such [bell-shaped] velocities non-uniquely (page 10)…Accordingly metabolic energy expenditure is a physiological, independently testable measure of effort.”

As pointed out in the paper, in minimum-variance theory, an effort term is implicitly there. This could indicate that minimum-variance theory is sufficient to explain all the data (in addition to the speed-accuracy tradeoff, which this work does not address). If this is incorrect, it would be useful for the authors to point out where minimum-variance theory falls short. For instance, recent work by Nagamori et al., 2021 questions the main assumption behind the theory.

We did not intend to dismiss the minimum-variance theory, and have amended the text to be clearer. In Discussion:

“However, the ambiguity also means that both variance and force-rate could potentially contribute to movement. It is quite possible that minimum variance dominates for fast and accurate movements, and energy cost for the trajectory and duration of slower ones, with both contributing to a unified objective for reaching.”

We agree with the reviewer that minimum-variance remains a viable theory, but it does not explain the metabolic cost predicted and measured here.